# Local Defogging Algorithm for the First Frame Image of Unmanned Surface Vehicles Based on a Radar-Photoelectric System

Qingze Yu *[iD] and Yumin Su

Science and Technology on Underwater Vehicle Laboratory, Harbin Engineering University, Harbin 150001, China; suyumin@hrbeu.edu.cn
* Correspondence: yuqingze@foxmail.com; Tel.: +86-181-0365-4056

**Abstract:** Unmanned surface vehicles frequently encounter foggy weather when performing surface object tracking tasks, resulting in low optical image quality and object recognition accuracy. Traditional defogging algorithms are time consuming and do not meet real-time requirements. In addition, there are problems with oversaturated colors, low brightness, and overexposed areas in the sky. In order to solve the problems mentioned above, this paper proposes a defogging algorithm for the first frame image of unmanned surface vehicles based on a radar-photoelectric system. The algorithm involves the following steps. The first is the fog detection algorithm for sea surface image, which determines the presence of fog. The second is the sea-sky line extraction algorithm which realizes the extraction of the sea-sky line in the first frame image. The third is the object detection algorithm based on the sea-sky line, which extracts the target area near the sea-sky line. The fourth is the local defogging algorithm, which defogs the extracted area to obtain higher quality images. This paper effectively solves the problems above in the sea test and dramatically reduces the calculation time of the defogging algorithm by 86.7%, compared with the dark channel prior algorithm.

**Keywords:** unmanned surface vehicle; radar-photoelectric system; fog detection; sea-sky line extraction; defogging algorithm

## 1. Introduction

An unmanned surface vehicle (USV) [1–4] is an intelligent surface boat that can perform maritime tasks through unmanned or remote control operations. USVs are more suitable for high-risk and repetitive maritime tasks than manned ships. Military and civilian applications of USVs are widespread.

USVs mainly use offshore defense, mine detection [5], and dynamic object tracking in the military. In the civilian, USVs are used mainly in hydrological monitoring, marine resource exploration, maritime search and rescue, and seabed exploration [6]. As shown in Figure 1a, the "Tianxing-1" USV is used for the experiment in this paper. Due to the special environment of the sea surface, USVs often encounter foggy weather when performing surface object tracking tasks, resulting in poor optical image quality and reduced target recognition accuracy. As shown in Figure 1b, it is a visual image of the "Tianxing-1" USV performing a maritime mission in a foggy environment.

### 1.1. USV Perception System

The USV perception system consists of a perception computer, a marine radar [7], and a photoelectric device. The perception computer includes object detection [8–11], object tracking [12–15], communication, and decision-making modules. The azimuth accuracy of the marine radar is 0.2 degrees, and the accuracy of the laser ranging is 5 m. As shown in Figure 2, firstly, marine radar detects the sea surface environment and obtains the approximate azimuth and distance of the object. Secondly, the object information is transmitted to the perception computer. The photoelectric sensor points the USV toward the

object according to the guidance of the perception computer. Third, the photoelectric device transmits the optical image to the perception computer after the target is guided. Then the perception computer obtains the target's pixel coordinates and category in the image through the object detection module. Fourth, the photoelectric device tracks the object in real-time according to the pixel coordinates of the object and the position and attitude information of the USVs and the photoelectric device. Fifth, the perception computer transmits the photoelectric video data with the detection results to the host computer on the shore through the radio station and displays it in real-time. Sixth, the host computer on the shore can send task instructions to the perception computer through a radio station.

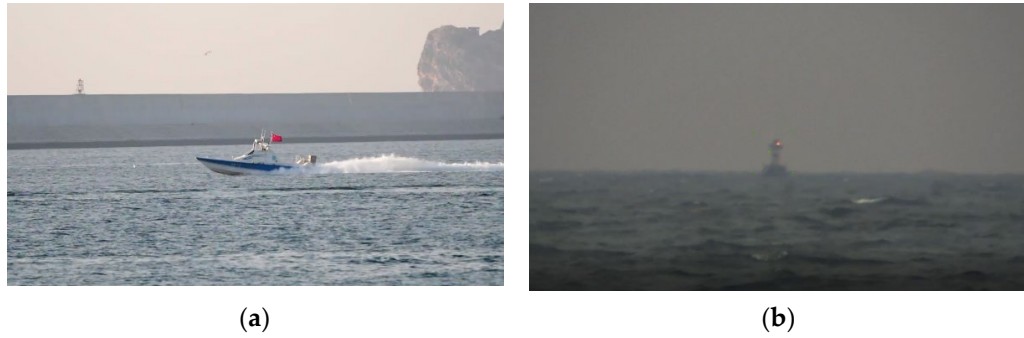

(**a**)                                  (**b**)

**Figure 1.** "Tianxing-1" unmanned surface vehicle: (**a**) The image of "Tianxing-1" during its mission; (**b**)The optical image of the "Tianxing-1" unmanned surface vehicle performing maritime missions in a foggy environment.

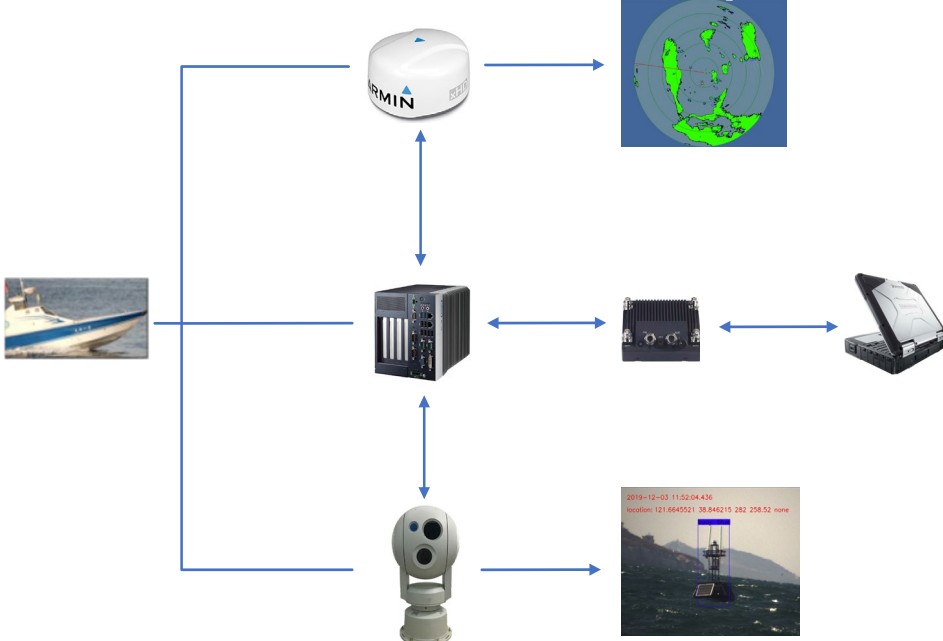

**Figure 2.** USV Perception system.

*1.2. Motivation and Contribution of the Current Work*

The current work is aimed at the following problems encountered in the object tracking of USVs using the navigation radar to guide the photoelectric mechanism: first, the fog on the sea surface causes the problem of low photoelectric visible light image quality. Second, when the traditional defogging algorithm is applied to the sea surface environment, the problems of over-saturated color, dark brightness, and halo are prone to occur. Third, the delay is too high when the traditional defogging algorithm is applied to real-time tracking. This paper proposes a defogging algorithm for the first frame image of the USVs based on the radar-photoelectric system. The first frame image refers to the first frame in the

sequence of images. The specific process is as follows: Firstly, the area with the USVs as the center and the radius of 2 km is detected by the marine radar, and the target's approximate azimuth angle and radial distance are obtained. Second, the marine radar transmits the echo information of the object to the perception computer, and through calculation and decision-making, the perception computer transmits the object's position information to the photoelectric device. Third, the photoelectric device points the object through the object position so that the object enters the field of view of the photoelectric device. Fourth, the photoelectric device transmits the image information of the object to the perception computer. Fifth, extract the first frame image after the object is in place. Sixth, extract the sea-sky line in the first frame image. Seventh, extract suspected objects near the sea-sky line. Eighth, extract the area near the object and perform image defogging to obtain higher quality images. In this way, the defogging algorithm of the first frame image of the USVs based on the radar-photoelectric system is realized. By predicting the object area, the algorithm removes the sky and ocean with relatively uniform colors so that the defogging algorithm optimizes the problems of color oversaturation, dark brightness, and halo. In addition, through the prediction of the object area, the calculation amount in the process of the defogging algorithm is significantly reduced, and the operation efficiency is improved, thereby meeting the real-time performance in the object tracking process of the USVs. The main contributions of this paper can be summarized as the following points:

1.  A fog detection algorithm for sea images is proposed

    According to the characteristics of the radar-photoelectric system's USV's first frame image, the algorithm compares and analyzes the foggy image and the fog-free image in the first frame image of the USVs. Through the brightness distribution characteristics of its grayscale histogram, the foggy condition of the sea surface image can be judged.

2.  A sea-sky line extraction algorithm based on a radar-photoelectric system for the first frame image of USVs is proposed

    According to the characteristics of the first frame image of the radar-photoelectric system, the algorithm counts the position of the sea-sky line in the first frame image of a large number of USVs and obtains the position area of the sea-sky line and the image characteristics near the sea-sky line. Through the two elements of predicting the area and the characteristic information of the sea-sky line, the sea-sky line information of the first frame image of the USVs of the radar-photoelectric system can be quickly obtained.

3.  A target area prediction algorithm based on the radar-photoelectric system for the first frame image of the USVs is proposed

    According to the characteristics of the first frame image of the USVs of the radar-photoelectric system, the algorithm performs statistics and analysis on the contrast characteristics of the sea target and the surrounding environment in the first frame image of the USVs. The distinguishing elements of the sea target and the surrounding environment are derived. This element achieves the purpose of quickly predicting the target area, reducing the extraction time by 86.7%, compared with the dark channel prior algorithm.

4.  A local defogging algorithm for the first frame image is proposed

    Through the prediction of the target area, according to the characteristics of the first frame local image of the USVs of the radar-photoelectric system, the algorithm performs statistics and analysis on the characteristics of the first frame partial image and the environmental image of a large number of USVs. In the environment, only the defogging operation is performed on the target area, which avoids the influence of the sky area on the defogging process and improves the defogging efficiency.

## 2. Related Work

The foggy environment poses a significant challenge to the regular and effective operation of many daily computer vision application systems. The existing image acquisition

equipment is susceptible to the interference of the external environment. In the foggy environment, the acquired images are often severely degraded, mainly manifested as blurred scene feature information, low contrast, and color distortion, which is not conducive to the computer vision system for the real image features. The extraction of the image will affect its subsequent analysis, understanding, recognition, and other processing series, which significantly reduces the practical application performance of the vision system and limits the application value of the image.

In essence, the purpose of image defogging is to remove the interference from weather factors of the degraded image and enhance the clarity and color saturation of the image. In this way, the valuable features of the image can be restored to the maximum extent so that the image can be better used in many computer vision systems such as remote sensing observation and automatic driving [16]. Therefore, it is of great practical significance to study how to effectively reconstruct the original clear image from the image captured in the foggy environment and improve the robustness of the visual system.

## 2.1. Defogging Algorithms Based on Image Enhancement

The defogging algorithm based on image enhancement does not consider the cause of image degradation, but improves the image's visual effect by enhancing the contrast. This kind of algorithm is widely used, but it may cause some loss or over-enhancement of the information of the prominent part. Whether the operation object is the whole image or the local area, the defogging algorithm can be divided into global and localized image enhancement based on image enhancement. Among them, histogram equalization transforms the grayscale histogram of the fog image into a uniform distribution form and increases the image contrast by increasing the pixel grayscale value range. For example, Stark [17] and Kim et al. [18] proposed an adaptive histogram equalization algorithm and a partially overlapping sub-block histogram equalization algorithm, respectively. Homomorphic filtering is a technique widely used in signal and image processing that combines grayscale transformation with frequency filtering to improve image quality. Retinex is a color vision model that simulates human perception under different lighting conditions. Based on this model, Adrian et al. [19] proposed a haze enhancement algorithm with remarkable effect. The image enhancement algorithm based on local variance determines the degree of image enhancement by calculating and comparing the size of the local standard variance and then performs local gray scale stretching. However, none of the above methods consider the essential cause of haze image degradation, so the enhancement effect is limited, and the robustness is often poor. Image enhancement and defogging technology usually takes the multi-scale information features of the original image itself as the main consideration and eliminates impurities in insensitive areas in the image by compensating for the image scene's contrast, brightness, and color saturation, to improve the visual effect of the image.

## 2.2. Defogging Algorithms Based on Image Restoration

The absorption and scattering of light cause the low visibility on foggy days by suspended particles in the atmosphere. By developing a mathematical model, the researchers have explained the imaging process and the included elements of foggy sky images. The model is first proposed by McCartney [20] based on atmospheric scattering theory and subsequently derived by Narasimha et al. [21]. The model believes that there are two main reasons for the degradation of the imaging results of the detection system under the strong scattering medium. First, the contrast decreases; second, ambient light such as sunlight is affected by the scattering of the medium in the atmosphere to form background light, and the two parts of light are superimposed to reach the imaging device, which affects the imaging effect. With this principle as a reference, many excellent defogging algorithms have been proposed. Such as, He et al. [22] proposed the dark channel defogging algorithm. Raanan [23] solves the scene transmission map according to the difference in color line distribution between haze-free images and foggy images and proposes a single-image defogging algorithm, which takes a long time and cannot be used to process images with

high fog concentration. Robby [24] proposed a cost function based on Markov random field to restore haze-free images according to different contrasts. Although it can obtain defogging images with a better visual experience, it often causes over-saturation and image distortion to occur. Meng et al. [25] proposed a regularized defogging method, which improved the accuracy of atmospheric transmittance by mining the inherent boundary constraints of the atmospheric transmittance function and achieved an excellent defogging effect. Berman et al. [26] proposed a method for the global evaluation of transmission maps based on non-local priors, which can simultaneously restore depth-of-field and fog-free images. However, when the atmospheric light is extreme, this method fails due to the inability to detect fog lines well. Zhu et al. [27] proposed a defogging method based on color attenuation prior, established a linear model according to the positive correlation between the fog concentration and the difference between image brightness and saturation, and learned the model parameters through supervised learning methods to restore scene depth information; this achieved single image defogging. Some studies based on dark channel prior are optimized in terms of algorithm efficiency and recovery accuracy.

### 2.3. Defogging Algorithms Based on Deep Learning

In recent years, convolutional neural networks [28] have attracted the attention of researchers as a representative algorithm of deep learning [29]. Because the network has the ability of representation learning, it can effectively capture the potential mapping relationship between the input signal and the output signal, and it shows good performance in the field of image processing. Convolutional neural networks usually consist of convolutional, pooling, and fully connected layers. The convolution layer uses the convolution operation to complete image feature extraction, and the pooling layer completes downsampling and reduces the dimension of the extracted features. Two adjacent layers form a convolution group, which is connected to the fully connected layer through several convolution layers. The fully connected layer realizes the classification of image feature information. After multi-layer convolution and pooling operations, the complexity of image processing problems with vast amounts of data is reduced. There are two types of defogging methods based on a neural network: one is a two-stage defogging algorithm, and the other is a one-stage defogging algorithm. The two-stage defogging algorithm relies on the atmospheric degradation model and uses the neural network to estimate the parameters in the model. Most of the early defogging methods are based on this idea. The one-stage defogging algorithm uses a neural network to directly restore the foggy input image to obtain a defogging image, often referred to as end-to-end defogging in deep learning. The two-stage defogging algorithm uses the neural network to obtain the transmission map or atmospheric light value in a regression way and also needs to combine the traditional prior evaluation. That is to say, the neural network is only a tool to obtain the atmospheric light or transmission map, and finally, it is still necessary to obtain a haze-free image according to the atmospheric scattering model. The single-stage neural network defogging algorithm is completely separated from the atmospheric scattering model. It does not need to evaluate the transmittance and atmospheric light a priori. Instead, it learns the mapping relationship between the foggy and haze-free images through training and obtains the result through the mapping relationship; a fog-free image. Nowadays, more and more researchers are inclined to use the single-stage defogging method. The representative ones include Cai et al. [30], who proposed a network called DehazeNet, which uses a neural network to estimate the atmospheric transmittance of the input image, and an atmospheric scattering model to obtain fog-free images. Li et al. [31] proposed a network called AOD-Net, which appropriately deformed the atmospheric scattering model formula and learned its related parameters through a neural network. Zhu et al. [32] proposed the DehazeGAN algorithm, which uses a method similar to AOD-Net to simultaneously estimate transmittance and atmospheric light values in a generative adversarial network. Ren et al. [33] proposed the MSCNN algorithm, which constructed sub-networks with different coarse and fine grains through a multi-scale network structure to achieve a rough and fine evaluation of

the transmission map. This algorithm effectively suppressed the halo phenomenon in the process of defogging. Mei et al. [34] completely abandoned the physical model, regarded the neural network as a black box, obtained the mapping relationship between the fog image and the fog-free image through training, and adopted the encoder-decoder joint residual network block structure to achieve the end-to-end performance.

## 3. Fog Detection Algorithm

During the mission of the USVs at sea, if it encounters foggy weather, the video image quality of the vision system will be significantly reduced. As a result, the USV cannot accurately complete the detection and tracking of the sea target, which seriously affects the autonomous navigation of the USVs.

Therefore, it is of great significance to improve the quality of video images through defogging algorithms. However, sea fog is not always present. In the absence of fog, performing a defogging algorithm on an image will seriously affect the quality of the original image. Therefore, autonomously identifying whether there is fog through images is of great significance.

The "Tianxing-1" USV collected data on sea targets in fog and non-fog, respectively. Through comparative analysis of experimental data, it is found that the contrast of foggy images is generally lower than that of non-fog images, and this feature is also manifested in the grayscale histogram. The brightness distribution of the images without fog has a wider range, and the brightness distribution of the images with fog is more concentrated, where the distribution in the range of 50 to 150 is extremely prominent. Using this feature, we can determine whether the visual image of the USVs is a fog image or not. As shown in Figures 3 and 4, the grayscale histogram distributions of the fog-free image and the fog image are shown.

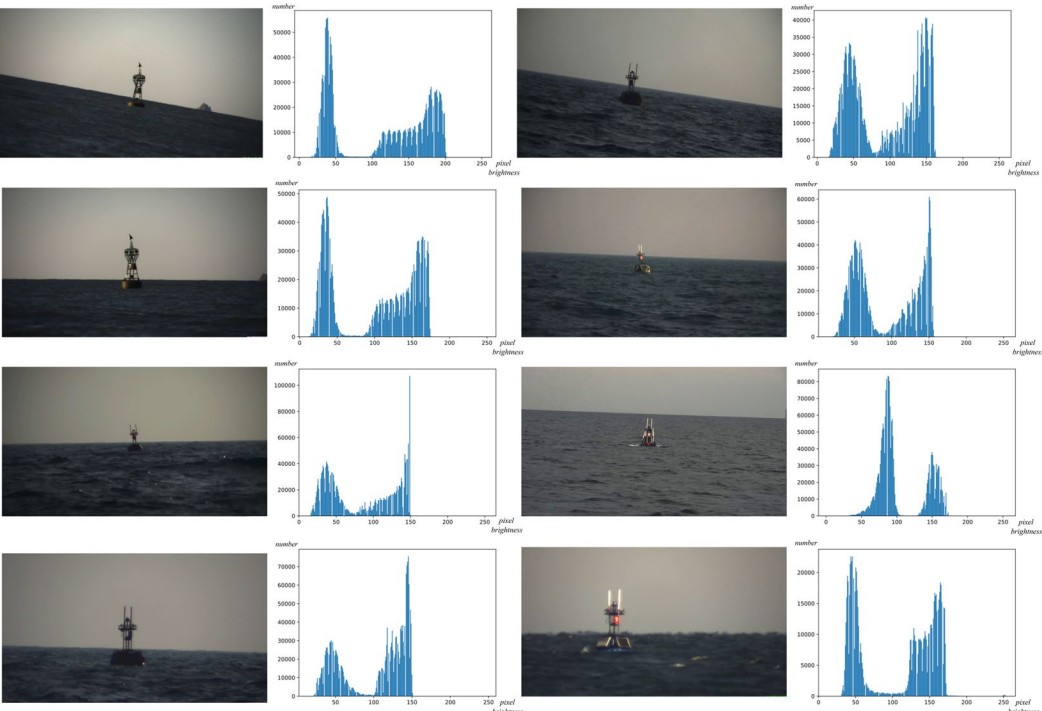

**Figure 3.** Fog-free image at sea and its grayscale histogram.

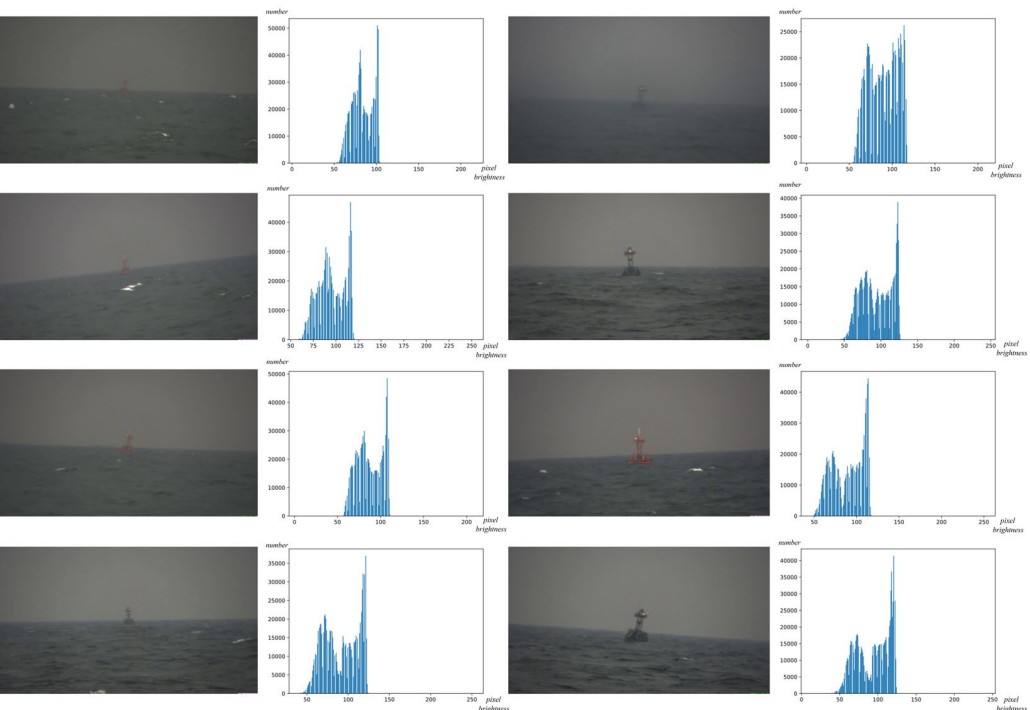

**Figure 4.** Foggy image at sea and its grayscale histogram.

A comparative analysis of the distribution characteristics of the grayscale histogram of the sea surface image with fog and the sea surface image without fog shows that the percentage $r$ of pixels with pixel brightness between $a$ and $b$ in the grayscale image with fog is higher than that of the grayscale image without fog. The image at sea is calculated based on the above properties.

$$n_{fog} = \sum_{i=a}^{b} n(i) \tag{1}$$

$$n_{total} = \sum_{i=0}^{255} n(i) \tag{2}$$

In the above formula, $n_{fog}$ is the number of pixels whose pixel brightness is between $a$ and $b$ in the grayscale image at sea. $n_{total}$ is the number of pixels with pixel brightness between 0 and 255 in the grayscale image at sea.

$$r = \frac{n_{fog}}{n_{total}} \tag{3}$$

$$c_{fog} = \begin{cases} 0, r \leq r_{fog} \\ 1, r > r_{fog} \end{cases} \tag{4}$$

In the above formula, $c_{fog}$ is the discrimination result, and $r_{fog}$ is the threshold for judging whether the image at sea is foggy. If $r$ is less than or equal to $r_{fog}$, then $c_{fog}$ is equal to 0, indicating that the sea surface image is a fog-free image. If $r$ is greater than $r_{fog}$, then $c_{fog}$ is equal to 1, indicating that the image at sea is foggy. As shown in Figure 5, it is the flow chart of the fog detection algorithm.

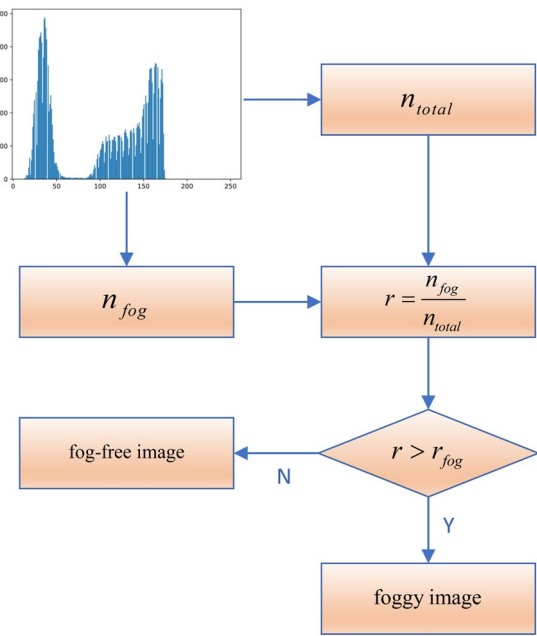

**Figure 5.** Flow chart of the fog detection algorithm.

## 4. Sea-Sky Line Extraction Algorithm

When the USVs perform various maritime tasks, the optical images obtained by its optoelectronic equipment have a common feature: the sky, the sea-sky line, and the sea surface. In addition, the target at sea is near the sea-sky line in the image obtained by the radar-photoelectric system. By extracting the sea-sky line, it is helpful to quickly determine the location area of the target, thereby reducing the computational complexity of image processing and improving the operation efficiency.

The sea-sky line divides the sky and the sea surface, and the images of the sky part are evenly distributed. The sea surface part has complex textures due to waves and sunlight reflection. In the edge extraction process, the sky part has less noise, the edge extraction of the sea-sky line is generally more complete, and the sea part has relatively more noise. Therefore, after extracting the edge, the bright spots can be screened from the top of the sky, and the noise in the sky can be filtered out by setting the brightness threshold, and then the maximum value is the position of the sea-sky line.

### 4.1. Preprocessing of Median Filtering

First, the image is preprocessed by a median filter, which can remove a lot of noise. At the same time, the main edge information in the original image can be retained to improve the accuracy of sea-sky line extraction. The median filter is a kind of sequential statistical filter in which the value of the pixel is replaced by the median gray value of the adjacent pixels.

$$g(x,y) = median\{f(x-i, y-j)\}, (i,j) \in S \tag{5}$$

In the above formula, $g(x,y)$ is the gray pixel value at $(x,y)$ after median filtering, and $f(x,y)$ is the central pixel gray value in the original image. $S$ is the template window, $i$ is the pixel difference in the horizontal direction, and $j$ is the pixel difference in the vertical direction.

### 4.2. Improved Sobel Edge Detection Algorithm

The edge of the photoelectric image is extracted by improving the Sobel edge extraction algorithm. Due to the appearance of fog, the adjacent pixels of the image at sea tend to have similar values, so the traditional Sobel edge extraction algorithm is very unsatisfactory for the foggy image at sea and cannot extract the edge features of the image well, as shown in Figure 6.

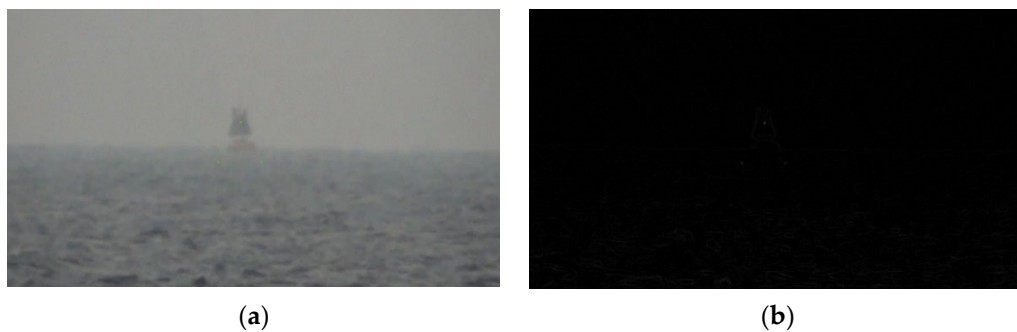

(a)　　　　　　　　　　　　　　　(b)

**Figure 6.** The extraction effect of the traditional Sobel edge detection algorithm: (**a**) A foggy image at sea; (**b**) The image extracted by the Sobel edge detection algorithm.

The improved Sobel edge detection algorithm in this paper removes a lot of redundant information for the characteristics of foggy images at sea and improves the original algorithm to obtain a better extraction effect. The specific algorithm is as follows:

Let $A$ be the image after the median filtering process, $g(x, y)$ be the gray value at $(x, y)$ in image $A$, and $s$ be the step size. According to the concentration of the fog, $s$ can choose parameters such as 2, 4, 8. When the fog is heavy, $s$ selects a large value to get a better effect.

$$h(x, y) = \frac{1}{s^2}[g(xs-1, ys-1) + g(xs-1, ys) + g(xs, ys-1) + g(x \cdot s, y \cdot s)] \tag{6}$$

Let $B$ be the image obtained from $A$ after the above calculation, $h(x, y)$ be the gray value at $(x, y)$ in image $B$, and $G_x$ and $G_y$ are the approximations of the gray bias in the horizontal and vertical directions, respectively. The mathematical expressions are as follows.

$$G_x = [h(x+1, y-1) + 2h(x+1, y) + h(x+1, y+1)] - [h(x-1, y-1) + 2h(x-1, y) + h(x-1, y+1)] \tag{7}$$

$$G_y = [h(x-1, y-1) + 2h(x, y-1) + h(x+1, y-1)] - [h(x-1, y+1) + 2h(x, y+1) + h(x+1, y+1)] \tag{8}$$

In the above formula, $h(x, y)$ is the gray value at $(x, y)$ in image $B$, from which $G_x$ and $G_y$ of each point can be calculated. For each point, we can obtain the gradient in two directions, and we can calculate the estimated value of the gradient by the following formula:

$$G = \sqrt{G_x{}^2 + G_y{}^2} \tag{9}$$

When $G$ is greater than the $G_{max}$ threshold, the point is white; otherwise, the point is black. Thus, an edge-detected image $C$ is obtained. Then perform bilinear interpolation calculation on the image $C$. The specific calculation is as follows:

$$h(x, y_1) \approx \frac{x_2 - x}{x_2 - x_1} h(p_{11}) + \frac{x - x_1}{x_2 - x_1} h(p_{21}) \tag{10}$$

$$h(x, y_2) \approx \frac{x_2 - x}{x_2 - x_1} h(p_{12}) + \frac{x - x_1}{x_2 - x_1} h(p_{22}) \tag{11}$$

$$h(x, y) \approx \frac{y_2 - y}{y_2 - y_1} h(x, y_1) + \frac{y - y_1}{y_2 - y_1} h(x, y_2) \tag{12}$$

where $p_{11}, p_{12}, p_{21}, p_{22}$ are the four adjacent pixel points with corresponding coordinates of $(x_1, y_1), (x_1, y_2), (x_2, y_1), (x_2, y_2)$. $(x, y)$ are the pixel points inserted in these pixel points. After the above operation, a higher quality edge detection image is obtained, as shown in Figure 7.

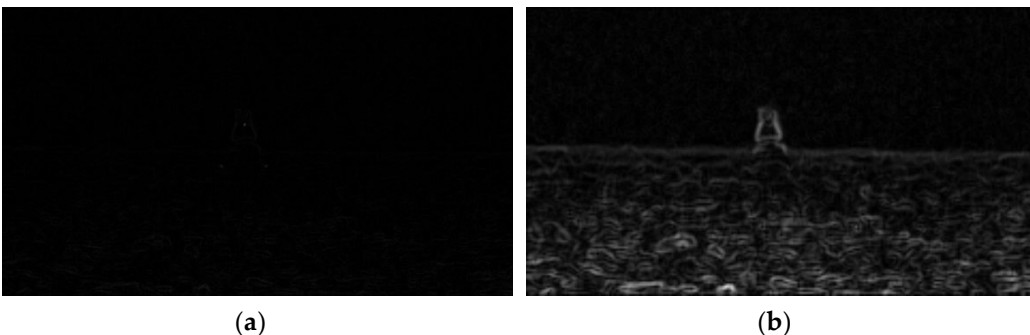

|     |     |
| :-: | :-: |
| (**a**) | (**b**) |

**Figure 7.** Extraction effect of improved Sobel edge detection algorithm: (**a**) traditional Sobel edge detection algorithm; (**b**) improved Sobel edge detection algorithm.

### 4.3. Sea-Sky Line Extraction Algorithm

By selecting the maximum value, the position of the sea-sky line is determined. After the image goes through the edge detection algorithm, there will be some low-brightness noise points in the sky. By filtering these noise points, the first bright spot in each column of the image is probably a point on the sea-sky line.

$$l(x,y) = \begin{cases} 0, h(x,y) \leq T \\ 1, h(x,y) > T \end{cases} \tag{13}$$

where $T$ is the brightness threshold, if $h(x,y)$ is greater than $T$, then $l(x,y)$ takes the value of 1, indicating that this pixel point is a bright spot; otherwise, the point takes the value of 0, which means it is filtered out.

$$m_i = F[l(i,1), l(i,2), l(i,3), \cdots, l(i,j_{\max})] \tag{14}$$

The specific practical effect is shown in Figure 8, where $m_i$ is the $j$ value of the first $l(i,j)$ in column $i$ with a non-zero pixel value.

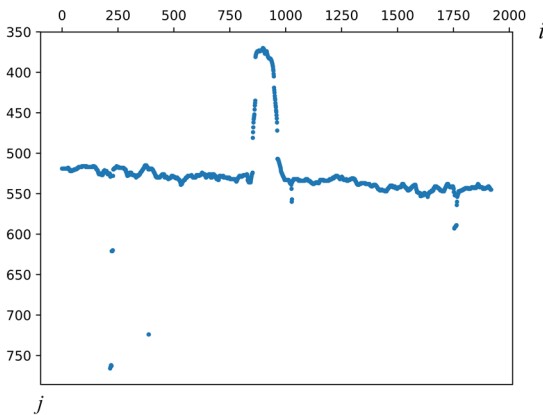

**Figure 8.** Scatter plot of the first bright point in each column.

The first bright point in each image's column is selected through the above method. Take these points as data and fit a line to them. The specific fitting process is as follows.

Let the sea-sky line equation be:

$$y = a + bx \tag{15}$$

$$\sum_{i=1}^{N} [y_i - (a + bx_i)] = \min \tag{16}$$

Take the partial derivatives with respect to $a$ and $b$, respectively:

$$\frac{\partial}{\partial a}\sum_{i=1}^{N}[y_i - (a + bx_i)]^2 = -2\sum_{i=1}^{N}(y_i - a - bx_i) = 0 \tag{17}$$

$$\frac{\partial}{\partial b}\sum_{i=1}^{N}[y_i - (a + bx_i)]^2 = -2\sum_{i=1}^{N}[y_i - (a + bx_i)]x_i = 0 \tag{18}$$

The best estimates of the line parameters $a$ and $b$ are obtained by solving the above equations:

$$\hat{a} = \frac{\sum_{i=1}^{N} x_i^2 \sum_{i=1}^{N} y_i - \sum_{i=1}^{N} x_i \sum_{i=1}^{N} x_i y_i}{N\sum_{i=1}^{N} x_i^2 - (\sum_{i=1}^{N} x_i)^2} \tag{19}$$

$$\hat{b} = \frac{N\sum_{i=1}^{N} x_i y_i - \sum_{i=1}^{N} x_i \sum_{i=1}^{N} y_i}{N\sum_{i=1}^{N} x_i^2 - (\sum_{i=1}^{N} x_i)^2} \tag{20}$$

According to the above method, the points in Figure 8 are fitted with a straight line, and the fitting result is shown in Figure 9. Figure 9a is the scatter fitting result; Figure 9b is the sea-sky line extraction result of the image after edge detection; Figure 9c is the sea-sky line extraction result of the original image. The red line in Figure 9 is the result of the sea-sky line extraction.

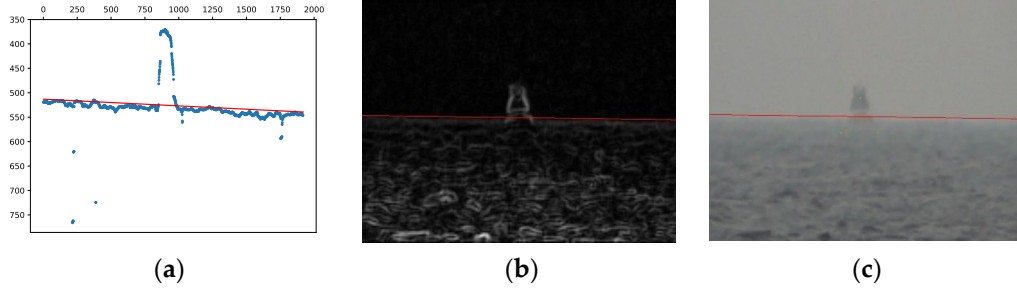

(**a**)　　　　　　　　　　　(**b**)　　　　　　　　　　　(**c**)

**Figure 9.** Image results of sea-sky line extraction: (**a**) scatter fitting result; (**b**) sea-sky line extraction result of the image after edge detection; (**c**) sea-sky line extraction result of the original image.

Due to the error caused by the object at sea and sea surface clutter, the fitting error of the sea-sky line extracted for the first time is relatively large. Therefore, the points extracted for the first time are filtered to reduce the fitting error of the sea-sky line. The specific operations are as follows.

$$\hat{a} + \hat{b}x - d \le y \le \hat{a} + \hat{b}x + d \tag{21}$$

For the points filtered by the above formula, perform linear fitting again, and the fitting result is shown in Figure 10. Figure 10a is the scatter fitting result; Figure 10b is the secondary sea-sky line extraction result of the image after edge detection; Figure 10c is the secondary sea-sky line extraction result of the original image. The red line in Figure 10 is the result of the first sea-sky line extraction, and the blue line in Figure 10 is the result of the secondary sea-sky line extraction.

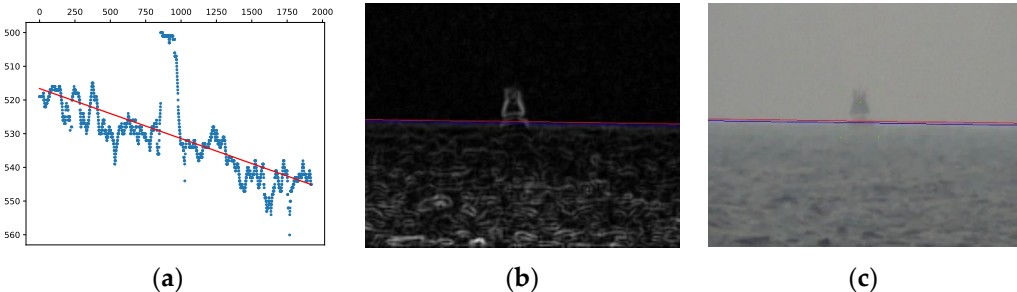

**Figure 10.** Image results of secondary sea-sky line extraction: (**a**) scatter fitting result; (**b**) secondary sea-sky line extraction result of the image after edge detection; (**c**) secondary sea-sky line extraction result of the original image.

## 5. Object Region Extraction and Local Defogging Algorithm

According to the perception system of the USVs and the characteristics of the image at sea, when the marine radar guides the photoelectric equipment to point to the object, the guided object will appear in the middle area of the optical image and near the sea-sky line at the same time. Therefore, the search of the object area only needs to be determined in the above area. Extracting the object area based on the sea-sky line can significantly reduce the amount of calculation, speed up the calculation speed, and improve the detection robustness.

As shown in Figure 11a, the semicircular window is slid along the sea-sky line and searched by step $s$, and the number of bright spots in the semicircular window at each position is counted. The specific calculation is as follows.

$$\begin{cases} \hat{a} + \hat{b}x \leq y \\ [x - (x_0 + n \cdot s)]^2 + \{y - [(\hat{a} + \hat{b} \cdot (x_0 + n \cdot s)]\}^2 \leq r^2 \end{cases} \tag{22}$$

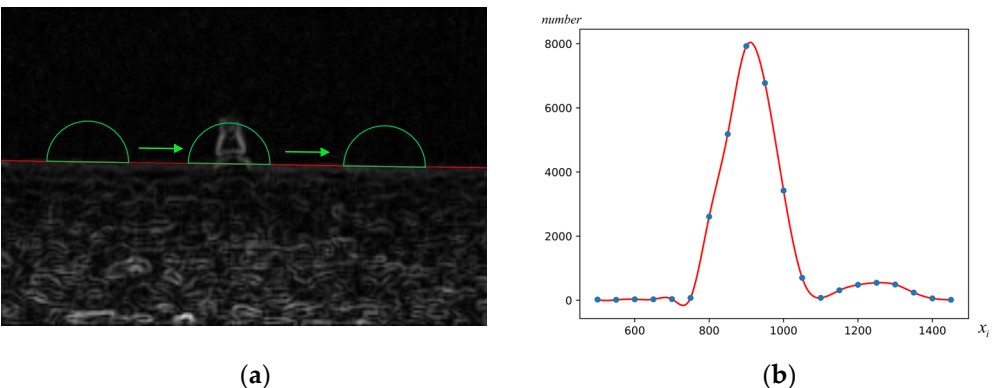

**Figure 11.** Target area extraction results: (**a**) target extraction process based on sea-sky line; (**b**) bright spot number curve based on sea-sky line.

The bright spots that satisfy the above formula are counted, and the distribution curve of bright spots near the sea-sky line can be obtained, as shown in Figure 11b.

The $x$ value corresponding to the maximum point in the curve is the $x$ coordinate of the center of the target area, and then fit a straight line according to the sea-sky line to determine the target area, as shown in Figure 12.

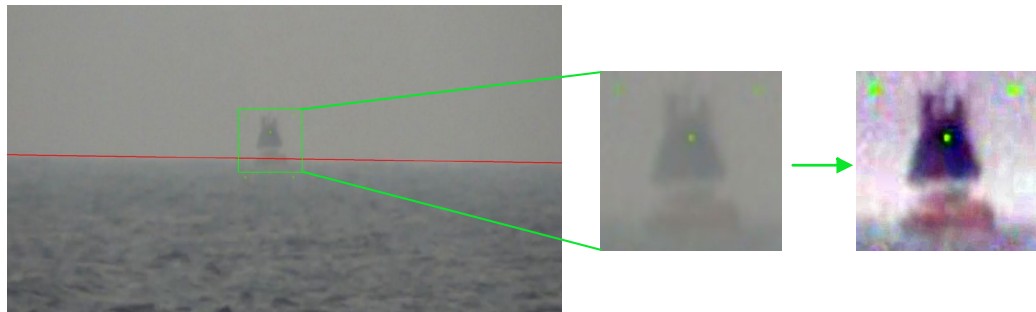

**Figure 12.** Schematic diagram of target area extraction and defogging of the area.

The target area in the foggy image can be extracted through the above method. The target in the target area is the focus of target recognition and detection, so it is enough only to perform the defogging operation in this area. In this way, the amount of calculation can be reduced, the dehazing effect can be improved, and the real-time performance of the defogging algorithm can be improved. The specific calculation process is as follows.

$$I(x,y) = J(x,y) \cdot t(x,y) + (1 - t(x,y)) \cdot A(x,y) \tag{23}$$

where $I(x,y)$ is the foggy image, $J(x,y)$ is the fog-free image, $A(x,y)$ is the global atmospheric light value, and $t(x,y)$ is the transmittance.

$$J^{\text{dark}}(x,y) = \min_{(x',y') \in \Omega(x,y)} \left( \min_{c \in \{r,g,b\}} J^c(x',y') \right) \tag{24}$$

where $\Omega(x,y)$ is a local neighborhood of pixel $(x,y)$, $c$ is a color channel, and $J^{\text{dark}}(x,y)$ is the dark channel of $J(x,y)$. The dark channel of the defog-free image in the non-sky region $J(x,y)$ tends to 0, meaning $J^{dark} \to 0$.

According to the atmospheric scattering model, Formula (12) is slightly processed and transformed into the following formula:

$$\frac{I(x,y)}{A(x,y)} = \frac{J(x,y)}{A(x,y)} \cdot t(x,y) + (1 - t(x,y)) \tag{25}$$

If $t(x',y')$ is assumed to be constant in the local neighborhood $\Omega$ of $(x,y)$, the following equation can be obtained.

$$t(x,y) \approx 1 - \min_{(x',y') \in \Omega} \left( \min_{c \in \{r,g,b\}} \frac{I^c(x',y')}{A^c(x',y')} \right) \tag{26}$$

$$J^c = \frac{I^c - A^c}{t} + A^c, \ c \in \{r,g,b\} \tag{27}$$

When $t(x,y) \approx 0$, the defogged image has the thickest defog at pixel $(x,y)$, meaning $J \to \infty$. Therefore, setting a minimum atmospheric transmittance $t_0$, the following equation can be obtained.

$$J^c = \frac{I^c - A^c}{\max\{t, t_0\}} + A^c, \ c \in \{r,g,b\} \tag{28}$$

When the algorithm calculates the whole image, it takes a lot of time, and it is difficult to ensure real-time performance. By extracting the target area, only the target area is calculated, which can reduce the calculation time by 86.7%, compared with the dark channel prior algorithm.

## 6. Conclusions

When performing surface object tracking tasks, USVs frequently encounter foggy weather, resulting in poor optical image quality and object recognition accuracy. Traditional

defogging algorithms are time consuming and inadequate for real-time applications. Furthermore, there are issues with oversaturated colors, poor brightness, and overexposed sky sections. This paper proposes a defogging algorithm for the first frame image of unmanned surface vehicles based on a radar-photoelectric system. The first is the fog detection algorithm for sea surface image, which determines the presence of fog. A comparative analysis of the distribution characteristics of the grayscale histogram of the sea surface image with fog and the sea surface image without fog shows that the percentage $r$ of pixels with pixel brightness between a particular interval in the grayscale image with fog is higher than that of the grayscale image without fog. Aiming at the problem that the traditional Sobel edge detection algorithm has, (a poor effect on foggy images), this paper improves it and obtains a better edge detection effect, thereby realizing the accurate extraction of the sea-sky line. A sea-sky line extraction algorithm is proposed, which employs a quadratic fitting approach, resulting in higher accuracy of the sea-sky line extraction results. The calculation amount is substantially decreased, the efficiency of the defogging algorithm is enhanced, and the time consumption of the defogging algorithm is lowered due to the suggestion of the target area extraction and local defogging algorithm based on the sea-sky line. By extracting the target area, only the target area is calculated, which can reduce the calculation time by 86.7%, compared with the dark channel prior algorithm.

**Author Contributions:** Conceptualization, Q.Y.; methodology, Q.Y.; software, Q.Y.; validation, Q.Y. and Y.S.; formal analysis, Q.Y.; investigation, Q.Y.; resources, Q.Y.; data curation, Q.Y.; writing—original draft preparation, Q.Y.; writing—review and editing, Q.Y.; visualization, Q.Y.; supervision, Y.S.; project administration, Y.S.; funding acquisition, Y.S. All authors have read and agreed to the published version of the manuscript.

**Funding:** This research received no external funding.

**Institutional Review Board Statement:** Not applicable.

**Informed Consent Statement:** Informed consent was obtained from all subjects involved in the study.

**Acknowledgments:** The authors are grateful to the anonymous reviewers and editors for their suggestions and assistance in significantly improving the manuscript.

**Conflicts of Interest:** The authors declare no conflict of interest.

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
