# Peer review of "Local Defogging Algorithm for the First Frame Image of Unmanned Surface Vehicles Based on a Radar-Photoelectric System"

_jmse, doi:10.3390/jmse10070969_

Round 1

Reviewer 1 Report

Congratulations for your work, I think that it's very good. But I have a general doubt:

In your paper, you detail the process and the differents improvements, but it's necessary include the processing time to evaluate if available in real time.

specific comments

From line 145 to 147. I don't understand the sentece, please could you check it? Then you explain better the differences between the two stages, but this sentence is confused.

In line 224. What is the time reduction in comparison with other systems 10 %, 90 %?

In section 2, Could you insert an graphical example of a,b, n_fog, n_total and r_fog calculation?

In lines 430 and 431, please quantify the reducing time.

The conclusions must be improve. Your work is good and you could explain more conclusions.

About Figures
Figures 3 and 4.:
    - Please introduce the figures before you show them.
    - It is necesasry improve the histogram ressolution, now I don'see clearly the axes values.
    - Where is the x label? What is the physical interpretation for x vales?
Figure 7 and 10b Please detail the labels to y and x axis.

Figure 8, 10. Please introduce the figure before you show it.

Figure 11. Your processing detail the target's area but the saturation images (more right in the figure) is from your processsing?

Reviewer 2 Report

The author's idea is quite interesting, but I have the following comments. That can further improve the quality of the manuscript.

1-Abstract is well written with background statements that support the current work such as why there is a need to do the following research work or what is the research gap. However, it would be better if some numerical results can be added to the last abstract. For example, how much percent the current scheme improves the results compared to the existing schemes.

2-Would be better if you can explain in one line, what you mean by the term “first image” just to give a clear picture to the reader. It’s a good idea to write the paper but think as a reader such as think readers know nothing about your domain, so you must make the things easier for them.

3-Just a suggestion, better is to always use the same tense in one paragraph. Otherwise, it makes things more complex. So, just review and refine the whole manuscript to remove the typo and grammatical mistakes and work on sentence structure.

4-I would strongly suggest you synchronize the paper heading, subheadings, captions, and table and inline the title of the paper as well.

4-In the abstract, the authors claim the existing works in literature were time-consuming and do not meet real-time requirements. So, what real-time requirements you have considered while doing the following research work, and what is the time complexity of your current algorithm that parts need to be highlighted a bit further.

5- I enjoy the paper flow, and the way use distributed the existing work into two subsections that’s why I would suggest that there should be a separate introduction and related work section to be more precise and clearer.

6- The heading of section 1.3 can be improved with Motivation and contributions of the current work or something like that instead of the content of this paper and highlights.

7-It looks like you have done a comparison between a fog-free image and its Grayscale image in Figures 3 and 4. However, if the quality of the plots is not up to the mark, I’m unable to see the values to see the output. You can improve the quality of the plots throughout the paper.

8-In your second proposed algorithm, what do you want to say in this line “In addition, the marine targets that the marine radar guides photoelectric detection are near the sea-skyline”. It is a bit confusing. And what are I and j in equation 5 discussing the medium filter formula?

9-How do you tune the step size for the second algorithm in equation 6?

10-Update the references in the paper with some recent work, it needs to be updated with the ongoing research and during the writeup of the paper.
